# Construction, Expression, and Evaluation of the Naturally Acquired Humoral Immune Response against *Plasmodium vivax* RMC-1, a Multistage Chimeric Protein

**DOI:** 10.3390/ijms241411571

**Published:** 2023-07-18

**Authors:** Ada da Silva Matos, Isabela Ferreira Soares, Barbara de Oliveira Baptista, Hugo Amorim dos Santos de Souza, Lana Bitencourt Chaves, Daiana de Souza Perce-da-Silva, Evelyn Kety Pratt Riccio, Letusa Albrecht, Paulo Renato Rivas Totino, Rodrigo Nunes Rodrigues-da-Silva, Cláudio Tadeu Daniel-Ribeiro, Lilian Rose Pratt-Riccio, Josué da Costa Lima-Junior

**Affiliations:** 1Laboratório de Imunoparasitologia, Instituto Oswaldo Cruz (IOC), Fundação Oswaldo Cruz (Fiocruz), Rio de Janeiro 21040-900, RJ, Brazil; adamatos.sm@gmail.com (A.d.S.M.); isaferreirasoares@gmail.com (I.F.S.); lana.bitencourt@hotmail.com (L.B.C.); 2Laboratório de Pesquisa em Malária, Instituto Oswaldo Cruz (IOC), Fundação Oswaldo Cruz (Fiocruz), Rio de Janeiro 21040-900, RJ, Brazil; bbaptista22@gmail.com (B.d.O.B.); hugoamaorims@gmail.com (H.A.d.S.d.S.); ericciovazoler@hotmail.com (E.K.P.R.); prtotino@ioc.fiocruz.br (P.R.R.T.); malaria@fiocruz.br (C.T.D.-R.); riccio@ioc.fiocruz.br (L.R.P.-R.); 3Laboratório de Imunologia Básica e Aplicada, Centro Universitário Arthur Sá Earp Neto/Faculdade de Medicina de Petrópolis (UNIFASE/FMP), Petrópolis 25680-120, RJ, Brazil; daiana.silva@prof.unifase-rj.edu.br; 4Laboratório de Imunologia Clínica, Instituto Oswaldo Cruz (IOC), Fundação Oswaldo Cruz (Fiocruz), Rio de Janeiro 21040-900, RJ, Brazil; 5Laboratório de Pesquisa em Apicomplexa, Instituto Carlos Chagas, Curitiba 81350-010, PR, Brazil; lesuaa@gmail.com; 6Laboratório de Tecnologia Imunológica, Instituto de Tecnologia em Imunobiológicos (Bio-Manguinhos), Fiocruz, Rio de Janeiro 21040-900, RJ, Brazil; rodrigo.nunes@bio.fiocruz.br; 7Centro de Pesquisa, Diagnóstico e Treinamento em Malária (CPD-Mal), Fiocruz e Secretaria de Vigilância em Saúde, Ministério da Saúde, Rio de Janeiro 21040-900, RJ, Brazil

**Keywords:** chimeric protein, *Plasmodium vivax*, protein construction, humoral immune response, B-cell epitope prediction, molecular modelling

## Abstract

The PvCelTOS, PvCyRPA, and Pvs25 proteins play important roles during the three stages of the *P. vivax* lifecycle. In this study, we designed and expressed a *P. vivax* recombinant modular chimeric protein (PvRMC-1) composed of the main antigenic regions of these vaccine candidates. After structure modelling by prediction, the chimeric protein was expressed, and the antigenicity was assessed by IgM and IgG (total and subclass) ELISA in 301 naturally exposed individuals from the Brazilian Amazon. The recombinant protein was recognized by IgG (54%) and IgM (40%) antibodies in the studied individuals, confirming the natural immunogenicity of the epitopes that composed PvRMC-1 as its maintenance in the chimeric structure. Among responders, a predominant cytophilic response mediated by IgG1 (70%) and IgG3 (69%) was observed. IgM levels were inversely correlated with age and time of residence in endemic areas (*p* < 0.01). By contrast, the IgG and IgM reactivity indexes were positively correlated with each other, and both were inversely correlated with the time of the last malaria episode. Conclusions: The study demonstrates that PvRMC-1 was successfully expressed and targeted by natural antibodies, providing important insights into the construction of a multistage chimeric recombinant protein and the use of naturally acquired antibodies to validate the construction.

## 1. Introduction

Malaria remains a significant public health threat, and the WHO estimates 247 million malaria cases globally in 2021. Between 2019 and 2021, there were an estimated 13.4 million cases [1]. While *Plasmodium falciparum* is the most prevalent malaria parasite in Africa and parts of Asia, *P. vivax* malaria remains the most widely distributed species that predominates in Central and South America, where the prevalence of *P. vivax* cases was almost 71.5% in 2021 [1,2]. In Brazil, 128.984 malaria cases were confirmed in 2022, 84% of which were due to *P. vivax*. Furthermore, severe, life-threatening *vivax* malaria is no longer considered a rare event [2]. The only licenced malaria vaccine to date, RTS,S, targets *P. falciparum* sporozoites but is not cross-protective against *P. vivax*. Therefore, to control malaria resurgence and make progress toward malaria elimination outside Africa, an effective vaccine targeting *P. vivax* is crucial.

In this scenario, the main strategies in malaria vaccine development are based on the *Plasmodium* biological cycle, with different antigens stimulating a stage-specific immune response against the pre-erythrocytic and intraerythrocytic, asexual and sexual parasite forms. The pre-erythrocytic strategy involves the generation of an antibody response capable of neutralizing sporozoites and then preventing hepatocyte invasion [3,4,5], generating a cell-mediated immune response to block the intrahepatic multiplication cycle by killing parasite-infected hepatocytes [6,7]; erythrocytic vaccines that prevent the development of the disease by inhibiting merozoite invasion and multiplication [8,9] through antibody-dependent cellular cytotoxicity [10,11] or complement lysis [12,13]; and transmission-blocking vaccines (TBVs) against the sexual stages of the parasite, preventing mosquitoes from becoming infective [14,15]. However, given the complexity of the *Plasmodium* spp. life cycle, an ideal antigenic composition for a potential vaccine could be targeted at multiple stages of the parasite. Thus, it becomes essential to explore new antigens to include them in this restricted list of vaccine candidates and, consequently, the use of the most immunogenic regions in chimeric constructs that can culminate in vaccines targeted at multiple stages of parasite development. In this context, three *P. vivax* proteins involved in different phases of the biological cycle were also explored for vaccine development: PvCelTOS (cell-traversal protein for ookinetes and sporozoites), PvCyRPA (cysteine-rich protective antigen) and Pvs25 (surface protein).

CelTOS (cell-traversal protein for ookinetes and sporozoites) is essential for the cell hepatocyte traversal of the malaria parasite in both mammalian and insect hosts; the mediation of transversal in mammalian hosts is a determinant of the success of malaria infection [16,17,18]. It is considered an attractive protein for immunizations because of its characteristic of being highly conserved among *Plasmodium* species [19], and vaccination with it can induce cross-reactive immunity against other *Plasmodium* species [20]. Regarding immunological aspects, PfCelTOS peptides have stimulated peripheral blood mononuclear cells (PBMCs) isolated from volunteers immunized with irradiated sporozoites, and this result was correlated with better protection of the volunteers, consolidating this protein as a potential vaccine candidate [21]. Additionally, immunization with PfCelTOS in mice demonstrated the ability to incite sterile protection against a challenge with *P. berghei* sporozoites in inbred and outbred mice [20], and mice immunized with recombinant PfCelTOS revealed inhibition of infection by a chimeric *P. berghei* parasite expressing the *P. falciparum* protein [22]. Moreover, this protein has been explored in other studies by our group, showing that PvCelTOS is naturally immunogenic and conserved in isolates of the Brazilian Amazon [23,24]. These results emphasized PvCelTOS as an antigen worth considering when designing a vaccine against *P. vivax*.

Corresponding to the erythrocytic stage, CyRPA (cysteine-rich protective antigen) plays a role in red blood cell invasion by the merozoite [25]. Due to the high polymorphism found in blood-stage candidates that is correlated with the ability to invade red blood cells by merozoites [26], the identification of new candidates has become an important focus of research in the malaria vaccine field. CyRPA belongs to a group of multiprotein complexes that are formed by the interaction of Ripr and RH5 (reticulocyte binding-like homologous protein) [27,28]. This protein complex colocalizes between merozoites and erythrocytes and binds to the erythrocyte receptor basigin, thus allowing invasion, and the CyRPA protein has a fundamental role in the formation of this complex [29,30,31]. However, a recent study showed that the basigin receptor is not considered essential for the invasion of *P. vivax* and belongs to a different protein complex. Nevertheless, CyRPA still has important functions during host cell invasion [32]. A study measuring IgG antibodies to 38 *P. vivax* antigens indicated that CyRPA was the protein that needed less antibody titration to be related to protection [33], reinforcing the important role of this potential vaccine candidate. In addition, our group also explored this protein and PvCyRPA, and despite the moderate sequence variation, the potential antibody targets did not seem to be significantly affected [34].

As a member of the P25 family of cysteine-rich 25 kDa antigens, Pvs25 has four tandem epidermal growth factor-like domains that are expressed on zygotes and mature ookinetes within mosquitoes [35,36,37]. The detection of low-level expression of P25 is possible in early gametogenesis, and an increase in its expression can be observed after fertilization [38]. Another point of interest is the fact that this protein is only expressed in the mosquito midgut, which is why it has not been under selective pressure by the vertebrate host immune system and presents less genetic variation [37,39,40,41], being also strongly conserved in Brazilian Amazon isolates [42]. Concerning the immunological data, mouse antisera to recombinant Pvs25 given together with *P. vivax* parasites completely prevented oocyst development in mosquitos [43]. Antibody levels obtained in a phase I vaccine trial of Pvs25 were correlated with transmission-blocking activity [44,45], and the same transmission-blocking activity correlation was observed in an immunization assay in mice and monkeys with its homolog protein of *P. falciparum* (Pfs25) [46]. Additionally, rabbit and monkey anti-Pvs25 antibody titers correlated with the reduction in the oocyst density per mosquito, and antibody levels also correlated with the mosquito infection rate [47]. In conjunction, these data highlight Pvs25 as a promising transmission-blocking vaccine candidate protein.

Here, we describe the construction and structural analysis of a multistage chimeric protein, confirming the antigenicity of its epitopes in a naturally exposed population of the Brazilian Amazon, as previously described in other scenario(s) [48].

## 2. Results

### 2.1. Structural Analysis of RMC-1

Using the ChimeraX program version 1.5 [49], we conducted 3D structural modelling of PvRMC-1 based on the translated amino acid sequence corresponding to the gene sequence used to express the recombinant protein. In this way, the three proteins that compose PvRMC-1 were demonstrated at different points of visualization (Figure 1A and Appendix A). The per-residue confidence colouring (pLDDT), a superposition-free score that evaluates the local distance differences of all atoms in a model [50], and the predicted aligned error (PAE) plot were also explored to determine the confidence of the protein prediction structure (Figure 1B,C). The presence of hydrogen bonds in the protein structure model was also demonstrated, as its presence is important for promoting stability [51], which is important when we think in a vaccine context (Figure 1D and Appendix A).

From this tool, we also explored the PvRMC-1 protein surface (Figure 2A), and we analysed the possible parameters of solubility (Figure 2B and Appendix A).

### 2.2. Epidemiological Profile of Studied Individuals

The study population was composed of 301 individuals living in three neighbouring municipalities of the Acre state in the Brazilian Amazon. In this population, the majority were adults, and all individuals were exposed to malaria infection throughout the year (Table 1). The age range was 12–91 years, with an average of 34.9 years. The cohort presented a nearly even gender distribution. The time of residence in malaria-endemic areas in the studied individuals ranged from 1.0 to 91.0 years, which indicated different degrees of exposure among the studied individuals. Concerning the prior history of malaria infections, 7% (*n* = 22) of all studied individuals did not remember previous malaria infections in their lives, and 3% (*n* = 8) reported no previous malaria episode during their entire life. Therefore, 90% (*n* = 271) of our studied individuals reported at least one malaria episode in the past. Among those who were positive at the time of blood collection (*n* = 100), the majority (81%) were diagnosed with *P. vivax* malaria. The number of past infections reported by individuals also varied greatly, ranging from 3 to 18 (mean = 12.04 ± 13.50), and the time elapsed since the last infection varied from 2 to 24 months (mean = 26.60 ± 46.08). Collectively, the evaluated parameters indicated that the studied population had different degrees of exposure and/or immunity.

### 2.3. Naturally Acquired Antibodies to Recombinant PvRMC-1 and the Prevalence of Cytophilic Antibodies in Brazilian Amazon Individuals

To verify whether the PvRMC-1 construct is targeted by the naturally acquired humoral response against *P. vivax*, we assessed the IgG reactivity profile against the chimeric recombinant protein using plasma samples from 301 individuals living in three endemic areas of the southwestern Amazon region of Brazil. First, we confirmed that PvRMC-1 was successfully expressed as an antigenic construct and was naturally recognized by antibodies from exposed individuals from the Brazilian Amazon. One hundred sixty-four (54.4%) individuals presented specific antibodies against PvRMC-1, with reactivity index (RI) values ranging from 1.00 to 8.80 (mean = 1.63 ± DP = 1.13) (Figure 3). Among these IgG responders, we observed a median to high prevalence of cytophilic IgG antibodies. There were 115 responders for IgG1 (70%) (median = 1.63; IQ = 1.29–2.17) and 113 responders for IgG3 (69%) (median = 1.39; IQ = 1.19–2.01). For the IgG2 subclass, 30 individuals were responders for the chimeric protein (18%) (median = 1.21; IQ = 1.05–1.49). We did not observe responders to the IgG4 subclass (Figure 4).

In a complementary way, to evaluate the acquired immune response to more recent stimuli, we evaluated the response against IgM antibodies. Of the 301 individuals, 119 (40%) were responders to IgM antibodies, with reactivity index (IR) values varying from 1.00 to 5.83 (median = 1.34 ± DP = 0.65) (Figure 3).

### 2.4. Association of The Specific Humoral Response against PvRMC-1, Clinical, and Exposure Factors

The frequency and reactivity index of IgG and IgM antibodies reactive to PvRMC-1 were significantly higher in individuals who were infected with *P. vivax* (IgG RI Median = 1.15, IQ range = 0.89–1.7; IgM RI Median = 0.89 IQ range= 0.70–1.37) and *P. falciparum* (IgG RI Median = 1.22, IQ range = 0.99–1.83; IgM RI Median = 1.21, IQ range= 0.80–1.85) at the time of blood collection when compared to individuals residing in the regions studied and who were negative both in microscopic and molecular diagnosis (IgG RI Median = 0.90, IQ range = 0.77–1.17; IgM RI Median = 0.80, IQ range = 0.64–1.10). However, we did not observe significant differences in the frequency and magnitude of response between those infected with *P. vivax* or *P. falciparum* (Figure 5A,B). The magnitude of response among the IgG responders was also maintained, with a predominance of cytophilic antibodies (IgG1 and/or IgG3) in the three groups of individuals studied (Figure 5C).

Regarding the correlation between epidemiological parameters and the IgG, IgM, and IgG subclasses, IgM was inversely correlated with age and time of residence in endemic areas, indicating that less exposed individuals have higher IgM levels than individuals with long-term exposure to malaria. On the other hand, IgG and IgM reactivity indexes were positively correlated with each other, and both were inversely correlated with MSLE. We could also not observe correlations between IgG subclasses and some of the explored epidemiological parameters, except for a correlation between the IgG2 subclass and age (r = 0.162; *p* = 0.037) and TREA (r = 0.209; *p* = 0.0153). Other correlations were observed among other subclasses: IgG3 with IgM isotype (r = 0.164; *p* = 0.035) and IgG1 (r = 0.224; *p* = 0.004), in addition to IgG3 and IgG4 (r *=* 0.205; *p* = 0.001) (Figure 6A,B).

### 2.5. Frequency of IgM Antibodies and IgG Subclasses among High and Low Responders

In a complementary way, we also evaluated the frequency of high and low responders among the 164 responders to IgG antibodies against chimeric PvRMC-1. Moreover, we evaluated the frequency of IgM and IgG subclass responses in each of these categories.

We considered high-responder individuals those with a reactivity index (RI) higher than 2, and we observed that 25 individuals (15%) were high responders, while 139 individuals (85%) were considered low responders (Figure 7A).

In the high-responder group, we found 16 (64%) responders to IgM, 14 (56%) responders to IgG1, 10 (40%) responders to IgG2, and 17 (68%) responders to IgG3. Among the group of low responders, we found 75 (54%) responders to IgM, 101 (73%) responders to IgG1, 20 (14%) to IgG2, and 96 (69%) to IgG3 (Figure 7B). We did not find any significant difference regarding the frequencies of responders for the IgM isotype between the two groups. In the low-responder group, (14) 10% of the individuals were considered infected for the first time (NPME = zero or 1), and in the high-responder group, only one individual (4%). Interestingly, we found a correlation between the IgG3 subclass and high-responder IR (r = 0.67; *p* = 0.003) and a correlation between low-responder individuals and MSLE (r = 0.230; *p* = 0.012).

## 3. Discussion

Over recent years, many advances have been made regarding the exploration of vaccine candidates targeting *P. vivax*. The approaches have three main aims: improved methods for evaluating vaccine potential before clinical trials (as *P. vivax* has no long-term in vitro culture), description of new vaccine candidates, and optimization of known vaccine candidates. In this study, we focused on the third goal and described the construction and expression of a chimeric protein, PvRMC-1, designed based on our previous works that described B-cell epitopes in conserved regions of PvCyRPA, PvCelTOS, and Pvs25. We also evaluated the acquired antibodies against this protein in a population naturally exposed to malaria in the Brazilian Amazon

First, in the design of PvRMC-1, we selected the full sequence of PvCyRPA after the signal peptide (255 amino acids) due to its small size and the moderate polymorphism found among field isolates in our previous work [34]. The two linear epitopes selected from Pvs25 were included as the main linear B-cell epitopes in strongly conserved regions of *P. vivax* isolates from endemic regions of the Americas and Asia [42,52,53]. In the C-terminal region, we also included PvCelTOS-derived TCD8 and the most antigenic linear B-cell epitope described in PvCelTOS [23,24]. With this design, we tried to construct a recombinant protein that contains at least one B-cell epitope from each of the sporozoite, merozoite, and gametocyte stages. Therefore, before the study of immunogenicity in animal models (in progress), we aimed to validate the recombinant chimeric protein using structure predictions and plasma samples from naturally exposed and infected individuals, confirming the maintenance of B-cell epitopes in the new recombinant that was successfully expressed.

Chimeric *Plasmodium* proteins have been explored for malaria vaccine development. For *Plasmodium falciparum*, a multiepitope chimeric antigen containing eight epitopes from this parasite has been explored. This antigen was well recognized by *P. falciparum* individuals in endemic regions along the China–Myanmar border, despite presenting cross-reactivity with *P. vivax*-infected individuals [54,55]. Furthermore, a chimeric antigen based on *P. vivax* MSP1 (PvRMC MSP1) was explored. This antigen contained an extended epitope present in variants of the MSP1 protein. The chimeric protein also showed an improved ability to recognize IgG antibodies from *P. falciparum*-infected individuals when compared to the recognition made by recombinant PvMSP1 [56]. More recently, McCaffery and collaborators explored the use of a chimeric protein composed of two proteins at different stages of the *Plasmodium* cycle: merozoite surface protein 1 (PvMSP1) genetically fused to *P. vivax* P25 (Pvs25), resulting in a recognition of native proteins that was greater for the chimaera than for the protein alone [56]. Based on these findings, allied to the complexity of the parasite cycle encompassing both vertebrate and mosquito hosts, [57,58] a vaccine representing different stages of the cycle is considered a promising tool, as explored in other studies [59,60,61,62,63].

The structural vaccinology field has become an important way of accelerating vaccine development, as demonstrated by the COVID-19 pandemic [64], not only with the identification of potential epitopes but also with the benefit of molecular modelling to analyse the potential for binding with host proteins and other important structural aspects [65,66,67]. Apropos, a recent study also employed various bioinformatics methods exploiting the structure of a multiepitope vaccine designed against *plasmodium* merozoites [68]. We also used a similar approach to indicate a possible exposition of epitopes on the surface of PvRMC-1. Our results revealed the presence of hydrogen bonds, giving stability to the chains, which is important according to other vaccinology studies, especially in constructed vaccine complexes [69,70]. Another observation was the hydrophilic predicted profile of the protein. It can thus foster interaction with water, facilitating its dilution with different adjuvants in future preclinical trial formulations [71]. On the other hand, the lack of a crystallographic structure of our chimeric antigen limited our intention to explore some structural aspects, such as the maintenance of B-cell epitopes on the surface; however, in this study, to partially overcome this limitation, we validated the antigenicity of our protein through seroepidemiological analysis using plasma samples from naturally exposed individuals.

In this scenario, the studied population included a majority of rainforest region natives and some migrants from nonendemic Brazilian areas who had lived in the endemic area for more than 10 years. The wide age range of these individuals, the variation in the number of previously recalled infections, and the time elapsed since the last malaria episode indicate differences in exposure and immunity. Moreover, the majority of individuals reported a prior experience with *P. vivax* and *P. falciparum* malaria, with the predominance of *P. vivax* infections; these findings are in agreement with findings in the literature [7,72]. Since the acquisition of clinical immunity against malaria is mediated by antibodies and depends on continued exposure to the parasite, this population was considered ideal for exploring the humoral response to PvRMC-1 and its possible relationship with indications of protection.

The PvRMC-1 construct was broadly recognized by naturally acquired antibodies from individuals living in the southwestern Amazon region of Brazil. We found that over half of the individuals (54.4%) had specific antibodies against PvRMC-1, indicating that the expressed antigen preserved the antigenicity of original epitopes and can be advanced to tests of immunogenicity in animal models. It is important to mention that the use of chimeric construction in *vivax* vaccine development has increased over the last few years, especially with erythrocytic antigens such as MSP-1 [60,73], RBP-1 [61], PvAMA [72], and others. In our study, in addition to confirming the antigenicity and maintenance of B-cell epitopes in our construction, the frequency of responders is compatible with the frequency found for the single proteins in previous work carried out in endemic areas of Brazil, where PvCelTOS had a frequency close to 20% [24], and from Asia, where PvCyRPA presented a frequency close to 25% in children [33]. It is also noteworthy that the IgG reactivity indexes for the chimeric antigen were also comparable to both proteins, which presented moderate to low reactivity indexes [24,33]. In a complementary way, to understand whether this IgG-mediated response presented an association with antibody subclasses, we also explored the profile of PvRMC-1-specific IgG1, IgG2, IgG3, and IgG4 antibodies. Cytophilic IgG subclasses are important since previous studies related to *P. falciparum* antigens have shown that immunity to blood-stage antigens depends on a specific pattern of immunoglobulin subclass response, in which IgG1 and IgG3 antibodies have a link with antibody-dependent cellular inhibition of parasite growth in vitro [74,75,76]. Moreover, cytophilic antibodies display a protective mechanism of killing the parasite in cooperation with blood monocytes (ADCI) [77,78]. In this study, we observed a prevalence of cytophilic IgG antibodies, specifically IgG1 and IgG3, among those who responded to the PvRMC-1 construct, which is common to other studies of chimeric and nonchimeric recombinant proteins, which were higher than 50% [24,79,80,81,82,83]. Cytophilic antibodies are also known to be involved in the clearance of infected red blood cells by phagocytic cells, suggesting that the humoral response detected against our chimaera could have a protective effect against *P. vivax* infection. Not only is IgG responsible for providing naturally acquired immunity against malaria, but it was also already demonstrated for *P. falciparum* that IgM antibodies could inhibit merozoite invasion and replication in red blood cells (RBCs) by the fixation and activation of complement on the surface of the parasite [84]. Moreover, this same study connected the rapid induction of the IgM response after the first infection in adult patients and its prominence during *P. falciparum* infection in children and adults with lifetime malaria exposure. For *P. vivax*, there are still few reports involving the IgM-mediated response, but it is already known that IgM is persistent in hypoendemic regions, and its levels remain unchanged during acute infection but increase significantly at recovery [85]. In addition, IgM is also considered a good serological marker of exposure when combined with IgG [86]. Considering the findings of our study, in which IgG and IgM showed correlations between reactivity indexes against PvRMC-1, and both levels also correlated with indications of exposure and a shorter time since the last episode to present antibodies against the chimeric protein, the joint action of these two immunoglobulin isotypes also suggests a synergistic effect in the immune response. Additionally, both IgG and IgM reactivity indexes were inversely correlated with MSLE, suggesting that a longer time interval since the last malaria episode is reflected in lower levels of IgG or IgM.

Finally, based on previous works that suggested that the humoral response against PvCyRPA, with low antibody levels compared with classical vaccine candidates, is related to protection, we also stratified IgG responders into response levels, subdividing them into high and low responders. The high frequency of low responders was expected since both proteins elicit moderate levels of antibodies in natural exposition [24,33]. Interestingly, the low responders to our chimeric antigen also had a longer time elapsed since the last malaria episode than the high responders. We can speculate that the high response to PvRMC-1 can indicate more recent exposure to malaria or that a low response can be more associated with protection. However, in both scenarios, the evaluation of immunogenicity and the functionality of anti-PvRMC-1 antibodies generated in preclinical trials are still needed.

## 4. Materials and Methods

### 4.1. Study Area and Volunteers

A cross-sectional cohort study was conducted involving 301 individuals from three different endemic areas of Acre: Cruzeiro do Sul (*n* = 155), Guajará (*n* = 67), and Mâncio Lima (*n* = 79). Samples from individuals living in nonendemic areas of Rio de Janeiro were considered the control group. Samples and survey data were collected from July 2018 to August 2018, coinciding with the period of increased malaria transmission in the Acre and Amazonas states. A high-risk area situated in the Juruá Valley in the state of Acre, where 99% of autochthonous cases are reported, includes Cruzeiro do Sul (07°37′50″ S/72°40′13″ W) and Mâncio Lima (07°36′49′″ S/72°53′47″ W). The municipality of Guajará (02°58′18″ S/57°40′38″ W) in the state of Amazonas, which is a medium-risk area, has also been included. In Brazil, the risk of contracting malaria is assessed using annual parasitological indexes (API). These indexes help classify transmission areas based on the number of autochthonous cases per 1000 inhabitants, categorizing them as high risk (≥50), medium risk (<50 and ≥10), low risk (<10 and >1), or very low risk (<1). Written informed consent was obtained from all donors, and the study was reviewed and approved by the Fundação Oswaldo Cruz Ethical Committee and the National Ethical Committee of Brazil. The investigations were carried out following the rules of the Declaration of Helsinki.

### 4.2. Epidemiological Survey

To evaluate the epidemiological factors that may influence the humoral immune response against the chimeric recombinant protein PvRMC-1, all donors were interviewed upon informed consent. The survey included questions related to personal exposure to malaria, such as years of residence in the endemic area, recorded individual and family previous malaria episodes, the use of malaria preventive measures, the presence/absence of symptoms, and personal knowledge of malaria transmission. Survey data were entered into a database created with the Epi-Info program version 3.4.3 (Centers for Disease Control and Prevention, Atlanta, GA, USA), and all epidemiological data were also stored in the Excel program for subsequent analysis.

### 4.3. Malaria Diagnosis and Blood Sampling

Peripheral blood samples were collected by venipuncture in heparin tubes. After centrifugation (350× *g*, 10 min), the plasma was collected and stored at −20 °C and transported to Immunoparasitology Laboratory, Fiocruz—RJ. Thin and thick blood smears of all donors were examined for malaria parasites. The parasitological evaluation was performed by examination of 200 fields at 1000 × 126 magnification under oil immersion, and a research expert in malaria diagnosis examined all slides. In order to enhance the sensitivity of parasite detection, molecular analyses were conducted on all samples. The process involved extracting DNA from the blood samples using the QIAamp DNA blood midi kit (Qiagen, Germantown, MD, USA) as per the manufacturer’s instructions. Polymerase chain reaction (PCR) was then carried out, employing specific primers for the genus (*Plasmodium* sp.) and species (*P. falciparum* and *P. vivax*), following a previously described method [87]. All donors found positive for *P. vivax* and/or *P. falciparum* in blood smears were subsequently documented and treated using the chemotherapeutic regimen recommended by the Brazilian Ministry of Health.

### 4.4. Structural Modelling

The 3D structural prediction and analysis of the biochemical characteristics of the chimeric recombinant protein PvRMC-1 were performed with the ChimeraX program version 1.5 (UCSF, California, CA, USA). The amino acid sequence was applied in the AlphaFold [88] using multiple commands in the Python computational language. With this approach, we also analysed solubility attributes and the presence of hydrogen bonds.

### 4.5. PvRMC-1 Design and Expression

The chimeric recombinant protein PvRMC-1 was expressed in *Escherichia coli* and produced from the previous immunological and molecular characterization of four T-cell epitopes and two B-cell epitopes from PvCelTOS [24], two linear B-cell epitopes from Pvs25 [42], and a 255 amino acid sequence containing both B- and T-cell epitopes of PvCyRPA, encompassing the entire protein. The protein was subsequently expressed and purified (Figure 8). The protein had a final concentration of 0.68 mg/mL, with a final volume of 2.94 mL, and the storage buffer used was 0.03% NLS. Additionally, the samples were lyophilized after the refolding test.

### 4.6. ELISA

Antibody binding to PvRMC-1 was evaluated in plasma samples by enzyme-linked immunosorbent assay (ELISA). Briefly, MaxiSorp 96-well plates (Nunc, Rochester, NY, USA) were coated with PBS containing 1.0 μg/mL recombinant protein. After overnight incubation at 4 °C, the plates were washed and blocked for 1 h at 37 °C. Individual plasma samples diluted 1:100 in PBS-Tween containing 2% BSA (Sigma-Aldrich, Saint Louis, MO, USA) were added to duplicate wells. After 1 h at 37 °C and three wash cycles with PBS-Tween, bound antibodies were detected with peroxidase-conjugated goat anti-human IgG (Sigma, St. Louis, MO, USA) diluted at 1:2000, followed by the addition of o-phenylenediamine and hydrogen peroxide. Optical density was identified at 490 nm using a SpectraMax 250 ELISA reader (Molecular Devices, Sunnyvale, CA, USA). The results for total IgG or IgM were expressed as reactivity indexes (RIs), which were calculated by the mean optical density of an individual’s tested sample divided by the mean optical density of 10 nonexposed control individuals’ samples, from Rio de Janeiro, plus 3 standard deviations. Subjects were scored as responders to PvRMC-1 if the RI of IgG and IgM against the recombinant protein was higher than one. Additionally, the RIs of IgG subclasses were evaluated in responder individuals by a similar method using peroxidase-conjugated goat anti-human IgG1, IgG2, IgG3, and IgG4 (Sigma, St. Louis, MO, USA). Reactivity indexes were calculated using OD from exposed individuals/OD from controls plus three SD. This formula is widely used in malaria seroepidemiological studies and is a high standard to guarantee the positivity of studied individuals. The means of the cut-off OD values were 0.178 for IgG1, 0.207 for IgG2, 0.105 for IgG3, and 0.180 for IgG4. For IgG and IgM, the cut-off OD values were 0.333 and 0.223, respectively.

### 4.7. Statistical Analysis

All statistical analyses were carried out using Prism 9.0 for Windows (GraphPad Software, Inc., San Diego, CA, USA). The one-sample Kolmogorov–Smirnoff test was used to determine whether a variable was normally distributed. The Mann–Whitney test was used to compare the RIs of IgG against recombinant PvRMC-1 between the studied groups. Differences in the proportions of the RI of IgG subclasses and epidemiological parameters were evaluated by Fisher’s exact test, and associations between antibody responses and epidemiological data were determined by the Spearman rank test. A two-sided *p*-value < 0.05 was considered significant.

## 5. Conclusions

This study offers significant contributions to the development and construction of a multistage chimeric recombinant protein using key epitopes from PvCYRPA, Pvs25, and PvCelTOS. Additionally, the study explores naturally acquired antibodies to validate this construction. These findings support the creation of innovative approaches for preventing and managing *P. vivax* infection. To further enhance the potential use of this antigen as a vaccine candidate, additional research is still required to evaluate the immunogenicity of PvRMC-1 in animal models.

## Figures and Tables

**Figure 1 ijms-24-11571-f001:**
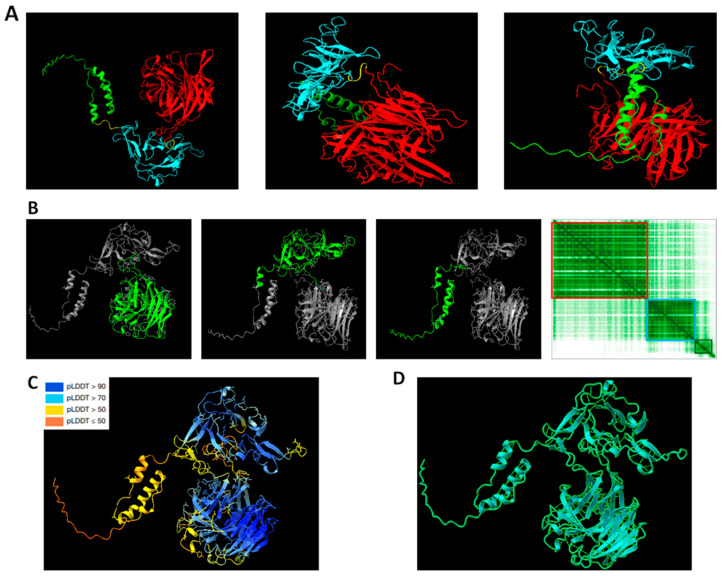
Predicted structure of PvRMC-1. (**A**) The chimeric protein is explored at different points of visualization; the red chain represents PvCyRPA, the blue chain represents Pvs25 epitopes, and the green chain represents PvCelTOS epitopes. (**B**) The red square in the PAE plot represents PvCyRPA, the blue is Pvs25, and the green is PvCelTOS. AlphaFold gives a distance error for every pair of residues X if the predicted and true structures were aligned on residue Y. The dark green squares in the plot represent a good confidence value. (**C**) Regions with pLDDT > 90 are expected to be modelled with high accuracy. These should be suitable for any application that benefits from high accuracies, such as characterizing binding sites. Regions with pLDDT between 70 and 90 are expected to be modelled well, and regions with pLDDT between 50 and 70 have low confidence. (**D**) The light brilliant green lines represent the hydrogen bonds, counted as 410.

**Figure 2 ijms-24-11571-f002:**
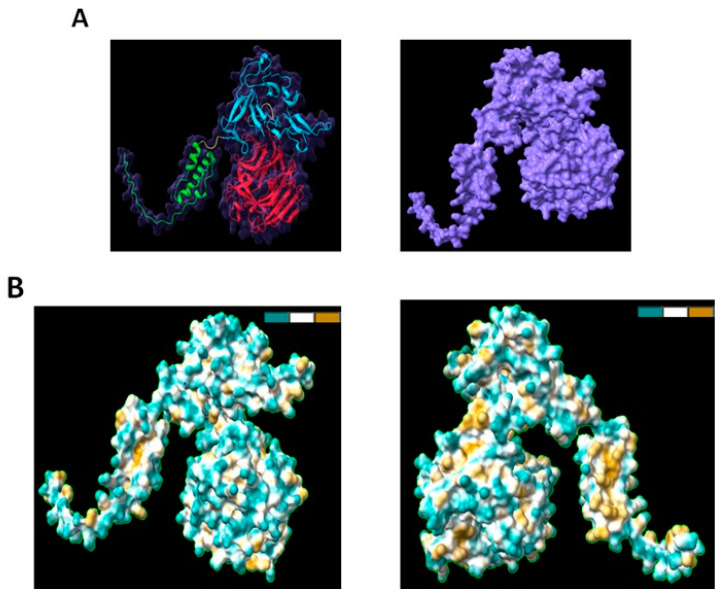
Surface structure and solubility parameter. (**A**) The complete surface of PvRMC-1: the red chain represents PvCyRPA, the blue chain represents Pvs25 epitopes, and the green chain represents PvCelTOS epitopes. (**B**) Visualization of its predominant hydrophilic characteristic. In ChimeraX, amino acid residues are automatically assigned an attribute named kdHydrophobicity, with values according to the hydrophobicity scale of Kyte and Doolittle. The hydrophilic profile is indicated in blue, neutral in white, and the hydrophobic profile, in dark yellow.

**Figure 3 ijms-24-11571-f003:**
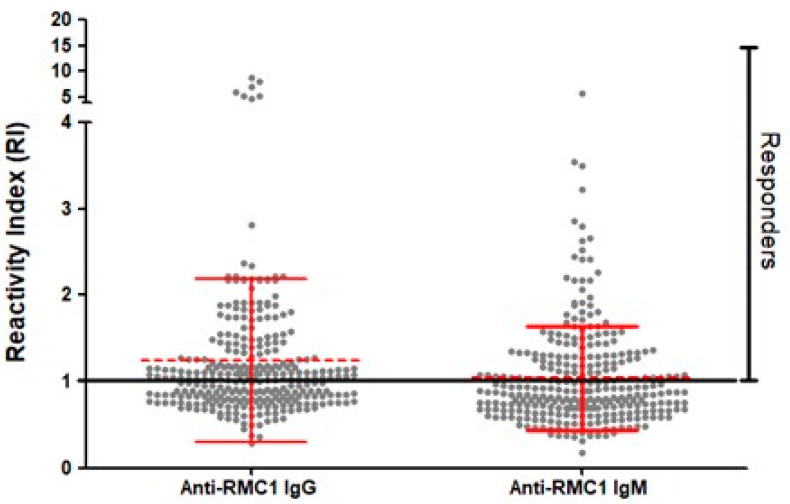
Naturally acquired antibodies to recombinant PvRMC-1. IgG and IgM reactivity indexes to the chimeric recombinant protein PvRMC-1 were evaluated in 301 individuals from a malaria-endemic region. Responders are represented above IR = 1 (black line). Means with SD are presented in red.

**Figure 4 ijms-24-11571-f004:**
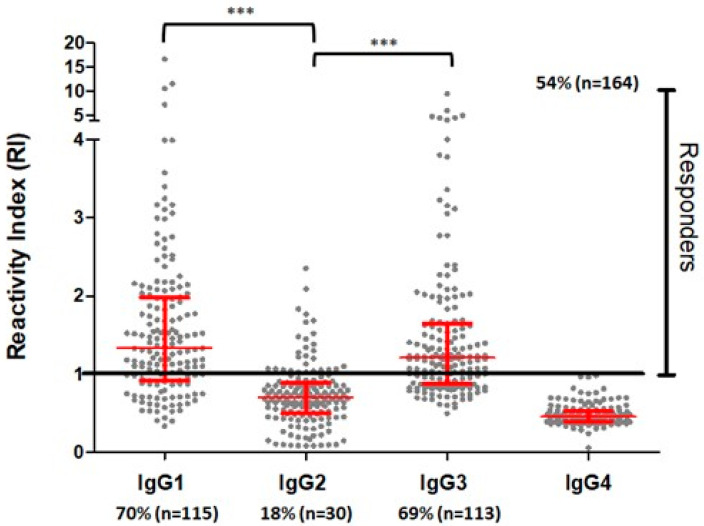
Frequency of responders to IgG subclasses and reactivity indexes of the population tested against the chimeric protein PvRMC-1. We evaluated the frequency of responders for IgG subclasses against PvRMC-1. There were no responders for the IgG4 subclass, and IgG1 and IgG3 frequencies were higher than that found for IgG2. Reactivity indexes of all individuals tested for IgG subclasses. The red line represents the median (IQ), and the black line represents responders for each subclass. Mann–Whitney *p* < 0.0001, represented as ***.

**Figure 5 ijms-24-11571-f005:**
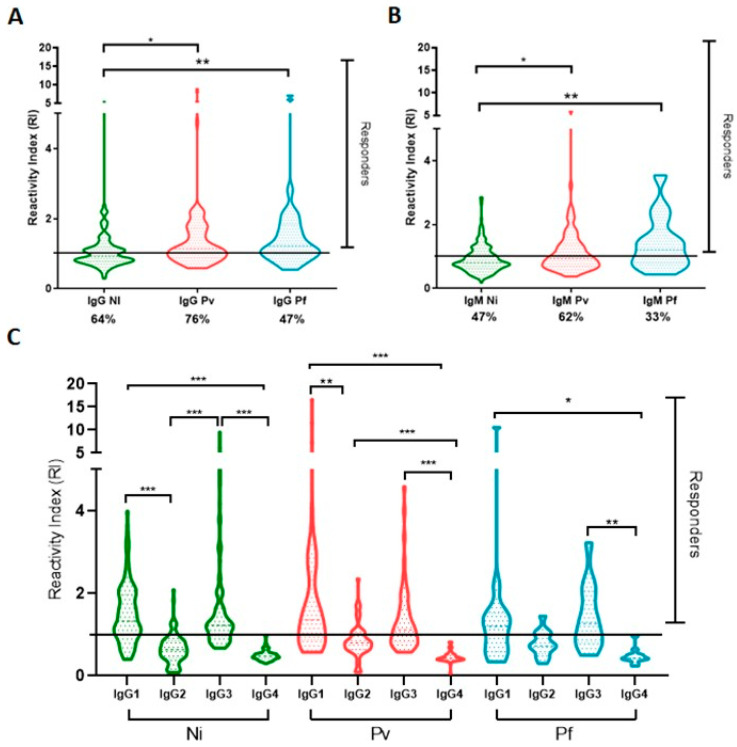
Reactivity index and frequencies related to exposure at the time of blood sampling. (**A**) IgG RI of *P. vivax*, *P. falciparum*, and noninfected individuals; (**B**) IgM of *P. vivax, P. falciparum*, and noninfected individuals. Frequencies were analysed using the chi-square test. (**C**) IgG subclass RI (IgG1, IgG2, IgG3, and IgG4) of *P. vivax, P. falciparum*, and noninfected individuals among PvRMC-1 responders. * *p <* 0.05; ** *p <* 0.005; *** *p <* 0.0005.

**Figure 6 ijms-24-11571-f006:**
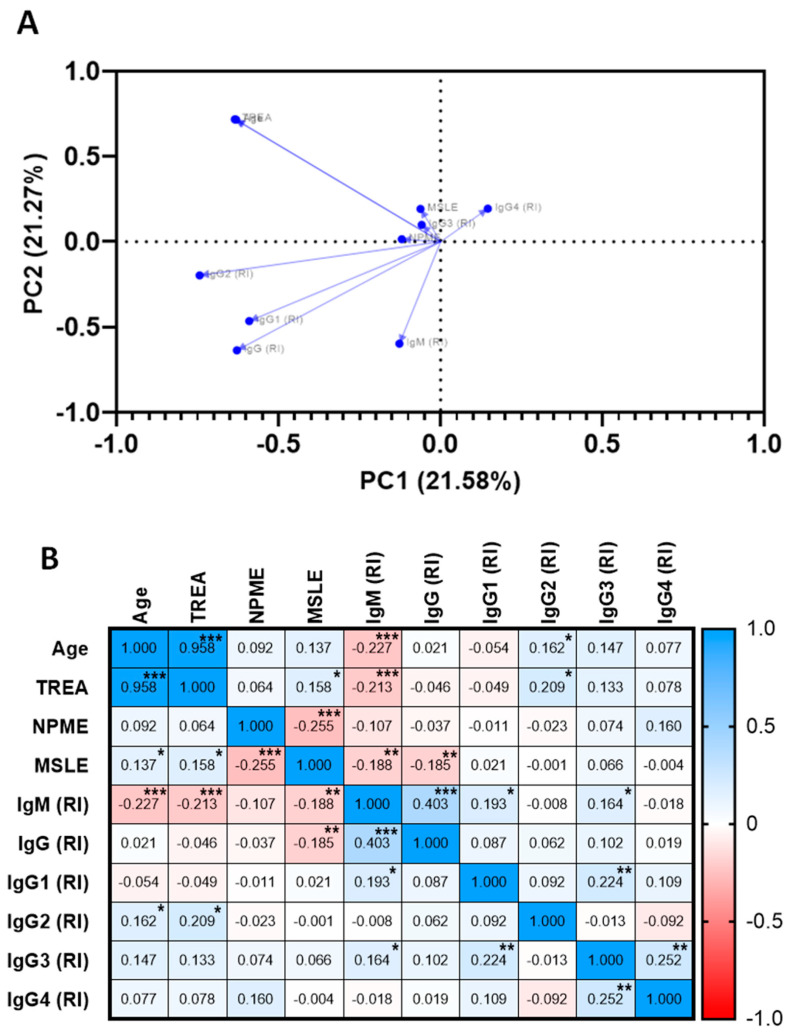
Association among specific antibody responses against PvRMC-1 and exposure variables in the studied population. (**A**) Principal component analysis of antibody response to PvRMC1 and epidemiological variables. Representation of principal component 1 (PC1 = 21.58%) and 2 (PC2 = 21.27%). The length of the arrows represents the strength in variability explaining each principal component related to the others. (**B**) Heatmap showing Spearman’s rank correlation between individuals tested for IgM, IgG, and subclasses with epidemiological parameters. The colours and intensity representing the coefficient of correlation (R values) ranged from blue (positive/direct coefficient of correlation) to red (negative/inverse coefficient of correlation). *p* values are expressed in boxes with statistically significant correlations (* *p <* 0.05; ** *p <* 0.01; *** *p <* 0.001).

**Figure 7 ijms-24-11571-f007:**
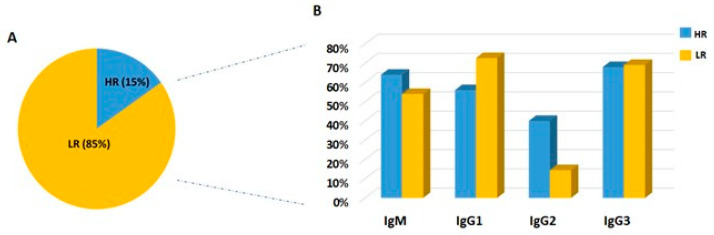
Frequency of high and low responders among the IgG responders. (**A**) The total representation of individuals is stratified into high responders (HR) and low responders (LR). Low responders were the majority when compared to high responders (*p* < 0.0001). (**B**) The blue bars represent the frequency of IgM and IgG subclass responders in the high-responder group (reactivity index > 2), and the yellow bars represent the frequency of IgM and IgG subclass responders in the low-responder group (reactivity index < 2).

**Figure 8 ijms-24-11571-f008:**
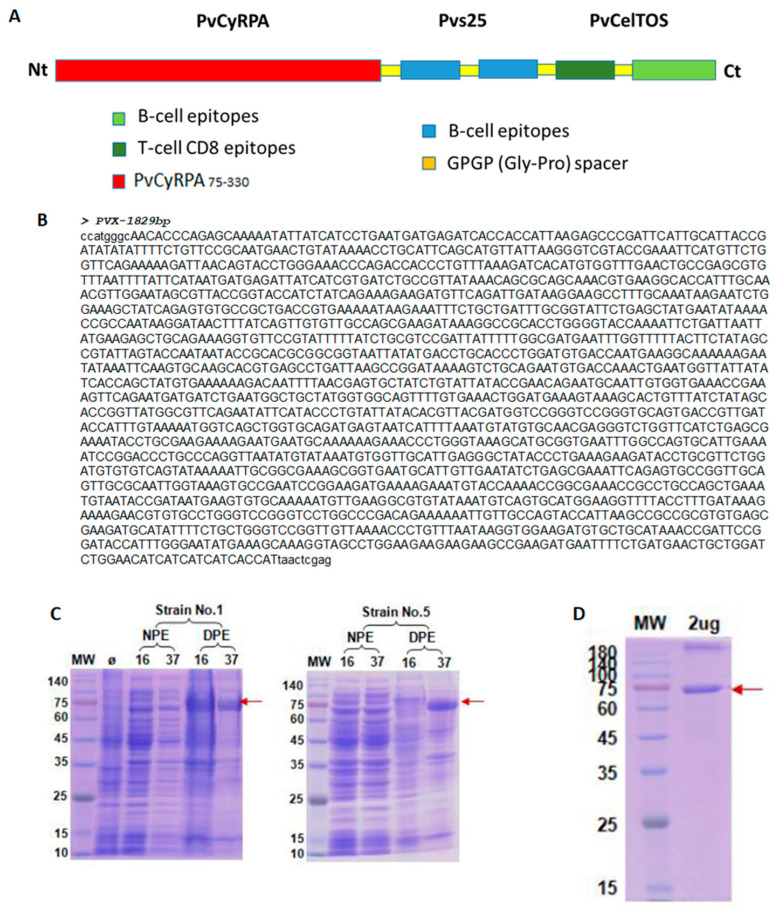
Chimeric protein composition, expression, and purification. (**A**) Composition of PvRMC-1 structure, with the predicted B-cell and T-cell epitopes; (**B**) gene coding: the cDNA was chemically synthesized with optimization for expression in *E. coli*; (**C**) SDS–PAGE analysis of recombinant protein expressed in two distinct *E. coli* strains stained with Coomassie brilliant blue: NPE (native protein extract), DPE (denatured protein extract), MW (molecular weight marker), Ø (noninduced bacteria culture), and 16 °C and 37 °C (incubation temperature); (**D**) SDS–PAGE analysis of recombinant protein after refolding, stained with Coomassie brilliant blue.

**Table 1 ijms-24-11571-t001:** Summary of the epidemiological characteristics of studied individuals enrolled in the survey. Values of age, time of residence in endemic areas (TREA), time of residence at the present address (TRPA), months since the last malaria episode (MSLE), number of malaria episodes in the last year, and number of previous malaria episodes (NPME) are represented by mean values (with standard variation). We also observed that the prevalence of *P. vivax* cases was higher than that of *P. falciparum* cases. The epidemiological parameters were compared using the Mann–Whitney test.

Epidemiological Features	Total (*n* = 301)
Male	50.5%
Female	49.5%
Malaria exposure	
Age—Mean (Sd)	34.86 ± 16.33
TREA *—Mean (Sd)	33.83 ± 16.35
TRPA *—Mean (Sd)	26.62 ± 18.09
Months since the last malaria episode— Median (IR)	9.0 (2.0–24.0)
Number of malaria episodes in the last year— Median (IR)	0.0 (0.0–1.0)
Number of previous malaria episodes— Median (IR)	8.0 (3.0–18.0)
Previous species contracted—N (%)	
*P. vivax*	58 (19%)
*P. falciparum*	21 (7%)
*P. vivax and P. falciparum*	192 (64%)
Never infected	8 (3%)
Diagnosis (PCR)—N (%)	
*P. vivax*	81 (27%)
*P. falciparum*	18 (6%)
Mixed	1 (0%)
Negative	201 (67%)

* TREA: time of residence in the endemic area; TRPA: time of residence at the present address.

## Data Availability

The datasets supporting the conclusions of this article are included within the article and its Appendix A.

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
