# Peer review of "Construction, Expression, and Evaluation of the Naturally Acquired Humoral Immune Response against Plasmodium vivax RMC-1, a Multistage Chimeric Protein"

_ijms, 2023, doi:10.3390/ijms241411571_

Round 1
Reviewer 1 Report
1. Improve the sentence “We performed 3D structural modelling of PvRMC-1 using the ChimeraX program 131 (50) based on the amino acid sequence sent for the subsequent synthesis of the protein.”’
2. P.vivax and all organism name should be in italics.
3. Author should provide detail on month of sample collection and also if it coincide with malaria season.
4. In Figure 4 how the threshold was decided. The same question is for Figure 5. I am asking since background of 1 od seems high.
5. Ca the author calculate the conc. of antibody in per ml of blood or plasma.
6. Can statstical test can be applied to data in Figure 8 ?
7. A linear diagram of RMC1 depicting domains of different proteins will be useful.
Moderate editing of English language required
Author Response
We were pleased with the overall positive feedback from the reviewers on our manuscript. We also appreciate the valuable comments and constructive suggestions received from the reviewers that helped to improve our manuscript. Below you will find an itemized list of the changes made in the revised version of the manuscript to address the specific concerns. We also like to apologize for any typographical errors, inconsistencies or misrepresentations in the original text that we have properly addressed in the revised version.
- Improve the sentence “We performed 3D structural modelling of PvRMC-1 using the ChimeraX program 131 (50) based on the amino acid sequence sent for the subsequent synthesis of the protein.”’
Answer: "Using the ChimeraX program version 1.5 (50), we conducted 3D structural modeling of PvRMC-1 based on the translated amino acid sequence corresponding to the gene sequence used to express the recombinant protein (Figure 1)."
2. P.vivax and all organism name should be in italics.
Answer: Thanks. We corrected all italics in text (highlighted in final version).
3. Author should provide detail on month of sample collection and also if it coincide with malaria season.
Answer: Thanks, the following sentence were included in material and methods section: Samples and survey data were collected from July 2018 to August 2018, coinciding with the period of increased malaria transmission in Acre and Amazonas States. A high-risk area situated at the Juruá Valley in the state of Acre, where 99% of autochthonous cases are reported, includes Cruzeiro do Sul (07º37′50’’S/72º40′13’’W) and Mâncio Lima (07º36′49’’S/72º53′47’’W). The municipality of Guajará (02º58′18’’S/57º40′38’W) in the state of Amazonas, which is a medium-risk area, has also been included. In Brazil, the risk of contracting malaria is assessed using Annual Parasitological Indexes (API). These indexes help classify transmission areas based on the number of autochthonous cases per 1,000 inhabitants, categorizing them as high risk (≥ 50), medium risk (< 50 and ≥ 10), low risk (< 10 and > 1), or very low risk (< 1).
4. In Figure 4 how the threshold was decided. The same question is for Figure 5. I am asking since background of 1 od seems high.
Answer: We did not show the OD values in graphs we showed the Reactivity index, which was calculated as described in Material and Methods (OD from exposed individuals/OD from controls plus 3 SD). This formula is widely used in malaria seroepidemiological studies and is a high standard to guarantee the positivity of studied individuals. However, taking into account your concern, we included all ELISA methods in the final version of the manuscript and decided to include the mean of cut-off OD values in the proper section to show the low background found in our assays.
5. Can the author calculate the conc. of antibody in per ml of blood or plasma.
Answer: Unfortunately, we only performed the ELISA with a plasma dilution factor of 1:100, which would not allow us to generate a standard curve using known concentrations of these samples for more robust analysis in this regard. Another point is that we do not have the availability to perform antibody purification in our laboratory. However, as described above we included all details of ELISA in the proper section in this final version.
6. Can statstical test can be applied to data in Figure 8?
Answer: Yes, we included the statistical significance between the overall IgG high and low responders in the figure legend. Unfortunately, the small number of individuals considered "high responders" limits us to further analyses with the IgG subclass.
7. A linear diagram of RMC1 depicting domains of different proteins will be useful.
Answer: Sure, thanks for this observation. We show a schematic representation of the proposed model in Figure 8A and subsequently performed 3D modeling of this protein, including the generation of videos. These approaches are the tools we have available to demonstrate the potential domains, as this was the only software capable of modeling a protein artificially constructed, considering commands generated by the AlphaFold program.
Reviewer 2 Report
This is an interesting study, given the difficulty of controlling malaria sustainably with conventional control tools. Thus vaccine development is crucial, particularly for Plasmodium falciparum. This study highlights the possibility of developing a vaccine against Plasmodium vivax, which is increasingly a threat in its traditional distribution areas.
Major comments:
Key information on malaria endemicity in the study areas is missing. It would be relevant to indicate the level of prevalence of the disease in the areas where the study population was recruited.
Also, in interpreting the results, the authors mention previous infections and even give the species of plasmodium responsible. How did they obtain this information? Was it based solely on surveys of the study population? If so, how reliable is this data?
Importantly, how were the participants selected? No indication is given of the inclusion process.
Of course, microscopic diagnosis is a reference technique, but is it sufficient to effectively diagnose malaria infection? We know that in some cases, it is very likely to find sub-microscopic infections. What impact can this have on the assessment of Ig response?
The authors have described the process by which PvRMC-1 is created, but in concrete terms they have given no information on the practical conditions for using this molecule: concentration of use, solubilization conditions.....
Similarly, no information is given on the techniques used to evaluate Ig. For example, did the authors use the Elisa technique or cytometry......?
The graphs show that a threshold is used to define positivity. How was this threshold calculated?
Author Response
We were pleased with the overall positive feedback from the reviewers on our manuscript. We also appreciate the valuable comments and constructive suggestions received from the reviewers that helped to improve our manuscript. Below you will find an itemized list of the changes made in the revised version of the manuscript to address the specific concerns. We also like to apologize for any typographical errors, inconsistencies or misrepresentations in the original text that we have properly addressed in the revised version.
1 - Key information on malaria endemicity in the study areas is missing. It would be relevant to indicate the level of prevalence of the disease in the areas where the study population was recruited.
Answer: We totally agree. The following sentence was included in Material and Methods section: Thanks, the following sentence were included in material and methods section: Samples and survey data were collected from July 2018 to August 2018, coinciding with the period of increased malaria transmission in Acre and Amazonas States. A high-risk area situated at the Juruá Valley in the state of Acre, where 99% of autochthonous cases are reported, includes Cruzeiro do Sul (07º37′50’’S/72º40′13’’W) and Mâncio Lima (07º36′49’’S/72º53′47’’W). The municipality of Guajará (02º58′18’’S/57º40′38’W) in the state of Amazonas, which is a medium-risk area, has also been included. In Brazil, the risk of contracting malaria is assessed using Annual Parasitological Indexes (API). These indexes help classify transmission areas based on the number of autochthonous cases per 1,000 inhabitants, categorizing them as high risk (≥ 50), medium risk (< 50 and ≥ 10), low risk (< 10 and > 1), or very low risk (< 1).
2 - Also, in interpreting the results, the authors mention previous infections and even give the species of plasmodium responsible. How did they obtain this information? Was it based solely on surveys of the study population? If so, how reliable is this data? Importantly, how were the participants selected? No indication is given of the inclusion process.
Answer: Yes. We agree with you on the limitations of our study design. The presence of P. vivax and P. falciparum infections in the majority of individuals examined suggests that both malaria parasites have been circulating in this region in recent decades. It is well-known that natural exposure to malaria infections is closely linked to the development of antibodies against Plasmodium spp. antigens. Our study population exhibited a wide range of ages, varied numbers of past malaria episodes, different durations of exposure, and varying months since the last reported malaria episode. These factors indicate that the inhabitants possess varying degrees of immune reactivity against PvRMC-1 protein components.
Due to the cross-sectional design of our study, we were limited to investigating retrospective malaria histories. The estimated time since the last malaria episode provided the best approximation of an individual's level of protection. However, ongoing prospective studies are being conducted to monitor the humoral immune responses over time, as well as biological studies that assess the ability of these antibodies to inhibit merozoite invasion or the development of blood-stage parasites. These studies will offer more direct evidence regarding the protective efficacy of the antibodies.
As suggested by you, we included the following information about the recruitment of volunteers: A cross-sectional cohort study was conducted involving 301 individuals selected from included by active case detection in three different endemic areas of Acre: Cruzeiro do Sul (n=155), Guajará (n=67) and Mâncio Lima (n=79).
3 - Of course, microscopic diagnosis is a reference technique, but is it sufficient to effectively diagnose malaria infection? We know that in some cases, it is very likely to find sub-microscopic infections. What impact can this have on the assessment of Ig response?
Answer: Sorry for this missing information. All samples were also evaluated by molecular techniques. We included the following information on the Diagnosis item in the Material and Method section: “In order to enhance the sensitivity of parasite detection, molecular analyses were conducted on all samples. The process involved extracting DNA from the blood samples using the QIAamp DNA blood midi kit (Qiagen, Germantown, MD, USA) as per the manufacturer's instructions. Polymerase Chain Reaction (PCR) was then carried out, employing specific primers for the genus (Plasmodium sp) and species (P. falciparum and Plasmodium vivax), following a previously described method (Snounou et al. 1996)”
4 - The authors have described the process by which PvRMC-1 is created, but in concrete terms they have given no information on the practical conditions for using this molecule: concentration of use, solubilization conditions.....
Answer: Perfect. We included the following information: “The protein had a final concentration of 0.68 mg/ml, with a final volume of 2.94 ml, and the storage buffer used was 0.03% NLS. Additionally, the samples were lyophilized after the refolding test.”
5 - Similarly, no information is given on the techniques used to evaluate Ig. For example, did the authors use the Elisa technique or cytometry......?
Answer: We really sorry about this. The complete ELISA technique (Item 4.6) that we used were missed in the first version. We updated the material and methods with the complete item and also included the Cut-Offs for all Igs tested:
4.6 Elisa: Antibody binding to PvRMC-1 was evaluated in plasma samples by enzyme-linked immunosorbent assay (ELISA). Briefly, MaxiSorp 96-well plates (Nunc, Rochester, NY, USA) were coated with PBS containing 1.0 μg/ml recombinant protein. After overnight incubation at 4°C, the plates were washed and blocked for 1 h at 37°C. Individual plasma samples diluted 1:100 in PBS-Tween containing 2% BSA were added to duplicate wells. After 1 h at 37°C and three wash cycles with PBS-Tween, bound antibodies were detected with peroxidase-conjugated goat anti-human IgG (Sigma, St. Louis) diluted at 1:2000, fol-lowed by the addition of o-phenylenediamine and hydrogen peroxide. Optical density was identified at 490 nm using a SpectraMax 250 ELISA reader (Molecular Devices, Sunnyvale, CA, USA). The results for total IgG or IgM were expressed as reactivity indexes (RIs), which were calculated by the mean optical density of an individual’s tested sample divided by the mean optical density of 10 nonexposed control individuals’ samples, from Rio de Janeiro, plus 3 standard deviations. Subjects were scored as responders to PvRMC-1 if the RI of IgG and IgM against the recombinant protein was higher than one. Additionally, the RIs of IgG subclasses were evaluated in responder individuals by a sim-ilar method using peroxidase-conjugated goat anti-human IgG1, IgG2, IgG3, and IgG4 (Sigma, St. Louis). Reactivity indexes were calculated using OD from exposed individu-als/OD from controls plus 3 SD. This formula is widely used in malaria seroepidemiolog-ical studies and is a high standard to guarantee the positivity of studied individuals. The mean of the cut-off OD values were 0.178 for IgG1, 0.207 for IgG2, 0.105 for IgG3 and 0.180 for IgG4.
The graphs show that a threshold is used to define positivity. How was this threshold calculated? We utilized the mean calculation combined with three times the standard deviation, setting a threshold of a final result above 1.0 for positive individuals. This parameter aligns with previous studies conducted by our group and is also supported by the existing literature. This formula was also described in the last version of the article.
Round 2
Reviewer 2 Report
The authors have considerably improved the quality of the manuscript.
However, the presentation of the results needs to be clarified:
- how the authors obtained their previous results on malaria infection. If these data were obtained from the epidemiological survey, how reliable are they? Did the participants provide documentation of malaria diagnosis with the result? This part is not clear, even in the methodology.
- How does the immune response evolve according to the geographical origin of enrolled individuals? Wouldn't it be more interesting to correlate this response with the annual parasitological index?
- Are the parasitological results in Table 1 obtained by microscopy or PCR?
Author Response
Dear Reviewer 2,
Thank you once again for your positive feedback and the detailed and insightful review of our manuscript.
Please find our responses below:
- how the authors obtained their previous results on malaria infection. If these data were obtained from the epidemiological survey, how reliable are they?
Answer: They were obtained by the survey and we included this information in the methods section in the revised version.
Using past malaria infections as reported in epidemiological surveys has limitations, especially regarding accuracy, which can reflect the lack of correlation found in our study. On the other hand, active case detection and cross-sectional study like ours, allow for individual-level analysis, enabling correlations with various factors such as age, immunity, genetic susceptibility, and treatment history. Thirdly, it serves as a proxy for exposure, considering not only the transmission intensity in a specific area but also individual behaviors and access to preventive measures. These advantages provide valuable insights into the impact of repeated infections on immune responses.
Did the participants provide documentation of malaria diagnosis with the result? This part is not clear, even in the methodology.
Answer: Yes. We included this information (also the treatment) in the revised version.
- How does the immune response evolve according to the geographical origin of enrolled individuals? Wouldn't it be more interesting to correlate this response with the annual parasitological index?
Answer: Thanks for this observation. While the API index is a measure of malaria transmission intensity at the population level, the number of past malaria infections at the individual level can serve as a proxy for the intensity of exposure to malaria. It takes into account not only the transmission intensity in a specific area but also individual factors such as behavior, travel patterns, and access to preventive measures. This individual-level exposure measure can be more informative for our main focus which was to investigate the correlation between malaria infection history and other variables such as the antibody levels. In contrast, the API index primarily reflects the current malaria transmission intensity and may not capture the individual's historical exposure to the disease. It is important to acknowledge that relying on the number of past malaria infections has limitations in terms of accuracy, particularly when considering malaria episodes that occurred during childhood. The absence of significant associations in our study findings may reflect the impact of this limitation.
- Are the parasitological results in Table 1 obtained by microscopy or PCR?
Answer: Thanks. It was done by PCR. We included this information in the table of revised version
